**www.cambridge.org/qrd**

Chaperonin; cryo-EM; conformational change; TRiC; ADP

**Corresponding author:**
Yao Cong;
Email: cong@sibcb.ac.cn

Mingliang Jin: Present address: Department of Biochemistry and Biophysics, University of California San Francisco, San Francisco, CA, USA.

Huping Wang: Present address: MRC Laboratory of Molecular Biology, Cell Biology Division, Cambridge, UK.

Mingliang Jin and Yunxiang Zang have contributed equally to this work

# The conformational landscape of TRiC ring-opening and its underlying stepwise mechanism revealed by cryo-EM

Mingliang Jin[1], Yunxiang Zang[1,2], Huping Wang[1] and Yao Cong[1,3] (ID)

[1]Key Laboratory of RNA Innovation, Science and Engineering, Shanghai Institute of Biochemistry and Cell Biology, Center for Excellence in Molecular Cell Science, Chinese Academy of Sciences, China; [2]The Research Center of Chiral Drugs, Innovation Research Institute of Traditional Chinese Medicine, Shanghai University of Traditional Chinese Medicine, Shanghai, China and [3]Key Laboratory of Systems Health Science of Zhejiang Province, School of Life Science, Hangzhou Institute for Advanced Study, University of Chinese Academy of Sciences, Hangzhou, China

## Abstract

The TRiC/CCT complex assists in the folding of approximately 10% of cytosolic proteins through an ATP-driven conformational cycle, playing a crucial role in maintaining protein homeostasis. Despite our understanding of ATP-driven TRiC ring closing and substrate folding, the process and mechanisms underlying TRiC ring-opening and substrate release remain largely unexplored. In this study, by determining an ensemble of cryo-EM structures of yeast TRiC in the presence of ADP, including three intermediate transition states, we present a comprehensive picture of the TRiC ring-opening process. During this process, CCT3 detects the loss of γ-phosphate and initiates with the dynamics of its apical protrusion, and expands to the outward leaning of the consecutive CCT6/8/7/5 subunits. This is followed by significant movements of CCT2, CCT4, and especially CCT1 subunits, resulting in the opening of the TRiC rings. We also observed an unforeseen temporary separation between the two rings in the CCT2 side, coordinating the release of the originally locked CCT4 N-terminus, which potentially participates in the ring-opening process. Collectively, our study reveals a stepwise TRiC ring-opening mechanism, provides a comprehensive view of the TRiC conformational landscape, and sheds lights on its subunit specificity in sensing nucleotide status and substrate release. Our findings deepen our understanding of protein folding assisted by TRiC and may inspire new strategies for the diagnosis and treatment of related diseases.

## Introduction

The group II chaperonin TRiC/CCT plays a crucial role in maintaining cellular protein homeostasis. Dysfunction of TRiC is closely linked to cancer and neurodegenerative diseases (Bassiouni et al., 2016; Jin et al., 2019b).TRiC assists in folding approximately 10% of cytosolic proteins, including essential cytoskeleton proteins tubulin and actin (both of which are obligate substrates of TRiC) (Llorca et al., 2000; Llorca et al., 1999; Waldmann et al., 1995), cell cycle regulator CDC20 (Camasses et al., 2003), and the G-protein signaling-related element Gβ (Plimpton et al., 2015). Additionally, several proteins involved in oncogenesis, including p53, VHL tumor suppressor, and STAT3 (Kasembeli et al., 2014; McClellan et al., 2005; Trinidad et al., 2013), as well as retinal developmental factors like transducin α and PEX7, are also substrates of TRiC (Hunziker et al., 2016; van den Brink et al., 2003).

TRiC is an ATP-dependent macromolecular machine that facilitates substrate binding and folding through ATP-driven conformational transitions (Reissmann et al., 2012). It consists of two identical stacked rings, each containing eight paralogous subunits (sharing ~25–40% sequence identity in the same organism) arranged in a specific order (Kalisman et al., 2012; Leitner et al., 2012; Zang et al., 2016; Zang et al., 2018). The hetero-oligomeric nature of TRiC is critical for its functionality, enabling complex substrate-binding modes in its open form (Joachimiak et al., 2014) and the differentiation of nucleotide usage during its ring closure and opening (Jin et al., 2019a; Reissmann et al., 2012). An individual TRiC subunit comprises three domains: an equatorial domain (E domain) that contains the ATP-binding site and forms intra- and inter-ring contacts, an apical domain (A domain) that comprises the interaction sites with target proteins, and an intermediate domain (I domain) that connects the other two domains (Ditzel et al., 1998; Klumpp et al., 1997; Waldmann et al., 1995). Many key structural elements of group II chaperonins involved in nucleotide consumption have been delineated. Notably, the nucleotide-sensing loop (NSL) located in the I domain monitors the presence of the γ-phosphate of ATP (Pereira et al., 2012; Zang et al., 2016). TRiC has been shown to exhibit subunit specificity in complex assembly, ATP consumption, and ring closure (Cong et al., 2012; Jin et al., 2019a; Joachimiak et al., 2014; Liu et al., 2023; Zang et al., 2018).

TRiC folds and releases substrate by undergoing conformation changes driven by ATP binding and hydrolysis. While its ATP-driven ring-closing mechanism and substrate-folding process have been extensively investigated (Gestaut et al., 2023; Han et al., 2023; Hunziker et al., 2016; Jin et al., 2019a; Liu et al., 2023; Llorca et al., 2000; Llorca et al., 1999; McClellan et al., 2005; Plimpton et al., 2015; Trinidad et al., 2013; Waldmann et al., 1995; Wang et al., 2023; Zang et al., 2016), the process and underlying mechanism of TRiC ring-opening and substrate release after ATP hydrolysis remains poorly explored. For example, although the structures of murine, bovine, and human TRiC-ADP have been previously reported (Cong et al., 2012; Llorca et al., 1998), the limited resolution and/or the capture of only one conformational state hinder a detailed understanding of the TRiC ring re-opening process and substrate release.

To capture the TRiC ring-opening process and reveal the underlying mechanism, we resolved an ensemble of cryo-EM structures of yeast TRiC in the presence of 1 mM ADP. Among the five resolved conformational states, three are unforeseen conformational states, likely representing intermediate states transforming from closed to open states. We find that CCT3 initiates the TRiC ring opening, potentially related to its NSL activity. This is followed by the outward tilting of CCT6/8/7/5, then propagated to CCT2/4/1. When CCT1 exhibits a pronounced outward tilting, the TRiC ring fully opens. Overall, we captured a more complete picture of the TRiC ring-opening process and revealed the stepwise mechanism of this process, providing a more complete picture of TRiC's conformational landscape. Our study enhances the understanding of the functional specificities of TRiC subunits and substrate release.

## Results

### Determination of an ensemble of cryo-EM structures of TRiC in the presence of ADP

In the ATP-driven conformational cycle of TRiC, after ATP-hydrolysis and the release of γ-phosphate, TRiC binds with ADP and re-opens its rings for substrate release from its chamber (Cong et al., 2012). To capture the intermediate states and understand the TRiC ring-opening mechanism, we determined an ensemble of cryo-EM structures of TRiC in the presence of 1 mM ADP. Under this condition, we resolved five cryo-EM maps, named TRiC-ADP-S1 through TRiC-ADP-S5, based on the progression of ring opening (Figure 1A–E). The resolutions of these maps are 4.75 Å, 5.64 Å, 6.80 Å, 8.50 Å, and 4.05 Å, respectively (Supplementary Figure S1). Notably, the fully open S5 state was most dominantly populated (63.3%), while the S4 state was least populated (1.5%, Figure 1F). As a control experiment, we also determined the cryo-EM map of yeast TRiC in the presence of ATP-AlFx, a nucleotide analog that has been used to mimic the ATP-hydrolysis transition state and can trigger TRiC ring closure (Gestaut et al., 2023; Han et al., 2023; Hunziker et al., 2016; Jin et al., 2019a; Liu et al., 2023; Llorca et al., 2000; Llorca et al., 1999; McClellan et al., 2005; Plimpton et al., 2015; Trinidad et al., 2013; Waldmann et al., 1995; Wang et al., 2023; Zang et al., 2016). The map was resolved at the resolution of 3.58 Å (termed TRiC-ATP-AlFx, Figure 1G and Supplementary Figure S2), exhibiting a both-ring tightly closed conformation of TRiC. We were able to assign the subunits in the obtained maps based on the detection of the inserted CBP affinity tag on CCT3 (for S1/S2/S3 state maps, Supplementary Figure S3A) and the longer CCT1 E-domain insertion feature (for S1/S2/S3 and TRiC-ATP-AlFx maps, Supplementary Figure S3B–D), along with a

conformation comparison of S5 with yeast TRiC-AMP-PNP also in the open conformation (Zang et al., 2016) (Supplementary Figure S4A), combined with the known subunit ordering of TRiC (Kalisman et al., 2012; Leitner et al., 2012; Zang et al., 2016; Zang et al., 2018). We then built a model for each conformational state except for the S4 map at lower resolution (Figure 1).

Among the five states, TRiC-ADP-S1 (6.4%) exhibits a conformation with both rings tightly closed (Figure 1A, F), similar to the closed TRiC-ATP-AlFx map (Supplementary Figure S4B). This suggests that when bound with ADP, a minor population of yeast TRiC can also close both rings. TRiC-ADP-S2, constituting 22.8% of the population, shows an observable conformational change in the A-domain helical protrusion of the CCT3 subunit (Figure 1B, F). As a result, an enlarged hole forms at the top of the dome of TRiC, making it appear slightly more open compared to the closed S1 state (Figure 1B). TRiC-ADP-S3, comprising 6.0% of the population, appears much more open than S2, seemingly in the middle of the transition from a closed to an open ring (Figure 1C, F). Interestingly, in this state, the E-domain between the two rings on the CCT2 side appears separated (Figure 1C), a phenomenon not previously reported. The least populated state, TRiC-ADP-S4 (1.5%), likely represents a transient intermediate state. It appears mostly open, with the CCT2/4/1 subunits displaying slightly distinct conformations between the two rings (Figure 1D, F). Moreover, the dominantly populated S5 state (63.3%) exhibits a conformation with both rings widely opened (Figure 1E, F). This conformation resembles that of yeast TRiC-AMP-PNP (Cong et al., 2012; Zang et al., 2016), and is also similar to that of the human TRiC-ADP, except for a slight shift of the A-domain of the CCT6 subunit (Supplementary Figure S4A). Furthermore, its dominant population distribution indicates that this wide-open TRiC-ADP-S5 state likely represents the most stable state in the presence of ADP.

It is noteworthy that, to the best of our knowledge, the conformations of the TRiC-ADP-S2/S3/S4 states have not been previously reported (Figure 1B–D). Furthermore, except for the least populated, highly transient S4 state, the conformations of the cis- and trans-rings in the other four states are essentially identical. This observation suggests that, aside from the very transit state, TRiC rings likely exhibit positive cooperativity during the ring-opening process (Figure 1B–D, F).

### TRiC ring-opening process occurs in a stepwise manner

To capture conformational transitions of TRiC during its ring-opening process, we conducted a pairwise comparison of the five TRiC-ADP maps, with each map low-pass filtered to 8 Å. We first compared the S1 and S2 states and observed that, in the S2 state, the density of the apical helical protrusion in the CCT3 subunit was absent (Figure 2A). Corroborating to this, the CCT3 apical protraction exhibited a significantly larger B-factor compared to the other subunits (Figure 2B). These findings collectively suggest greater dynamics in this region. Additionally, the absence of this density distorts the original contact between the iris and constrains the apical helical protrusions of neighboring subunits. As a result, the A- and I-domains of CCT3 exhibit an outward tilt from the S1 to the S2 state, with the A-domain tilted up to 16 Å (Figure 2C). Overall, these dynamic motions of CCT3 appear to initiate the opening of TRiC's iris. We therefore propose that CCT3 serves as the first subunit to initiate TRiC ring opening.

Further structural comparison between the S2 and S3 states suggests that, in the S3 state, the A-domain of CCT3 loses contact with its neighboring subunits and appears mostly disordered

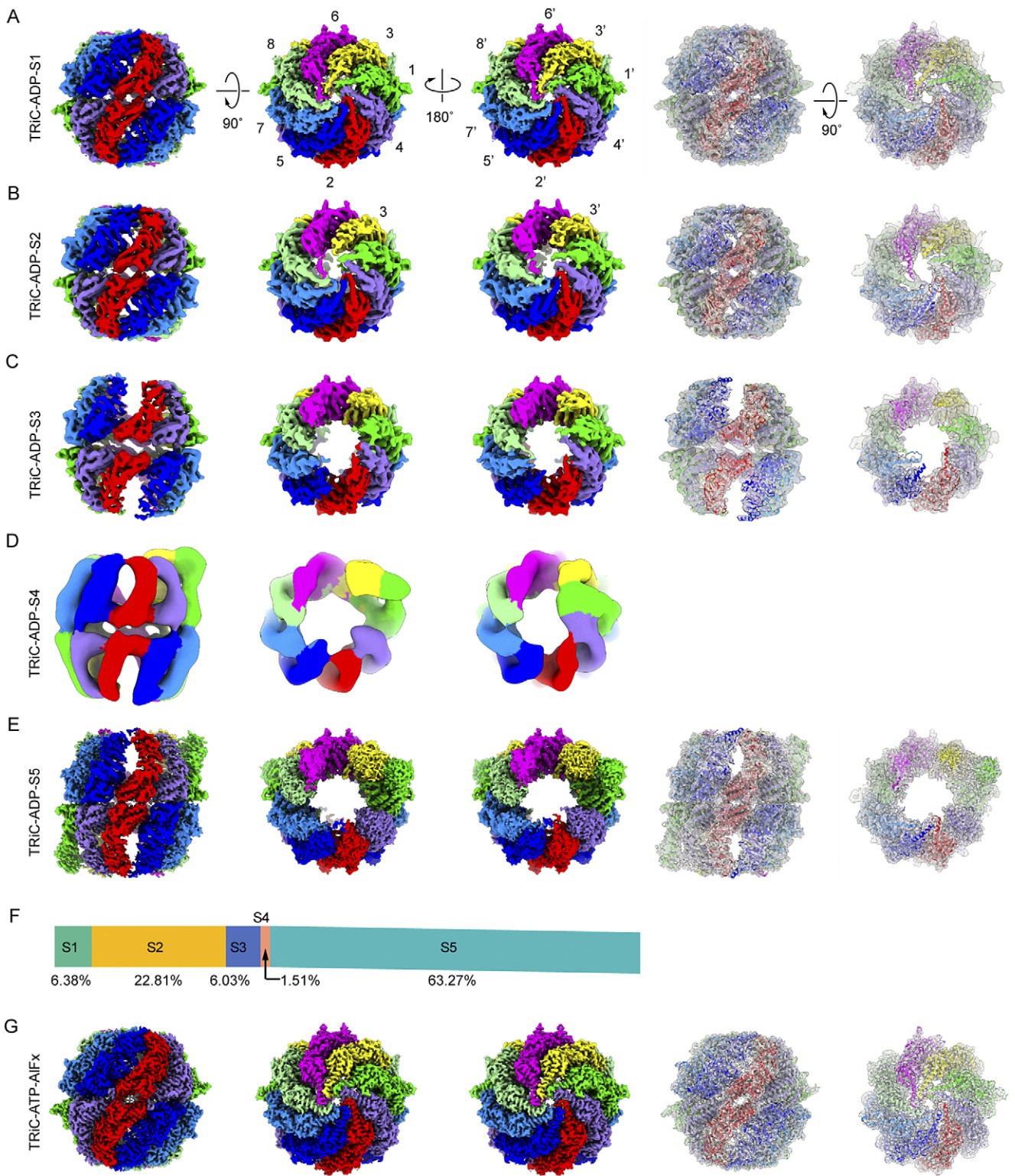

**Figure 1.** Cryo-EM structures of TRiC in the presence of 1 mM ADP. (A-E) Cryo-EM structures of yeast TRiC-ADP showing gradual ring opening. Models fitted into the maps are depicted on the right, except for TRiC-ADP- S4. (F) Population analysis of these TRiC-ADP maps. (G) Control TRiC-ATP-AlFx structure for comparation and analysis. The model fitted into the map is shown on the right.

(Figure 2D–E). Meanwhile, its counter-clockwise neighboring sub-units, including CCT6/8/7/5, expand outward in a coordinated manner, with the apical helix retracting up to 18 Å (Figure 2D, F, Supplementary Movie 1). This leads to a partial opening of both rings of TRiC in the S3 state (Figures 1C and 2D). In the meanwhile, the remaining three subunits, CCT2/4/1, also exhibit a slight out-ward tilt due to the loss of interaction in the A-domain with the other subunits (Figure 2D).

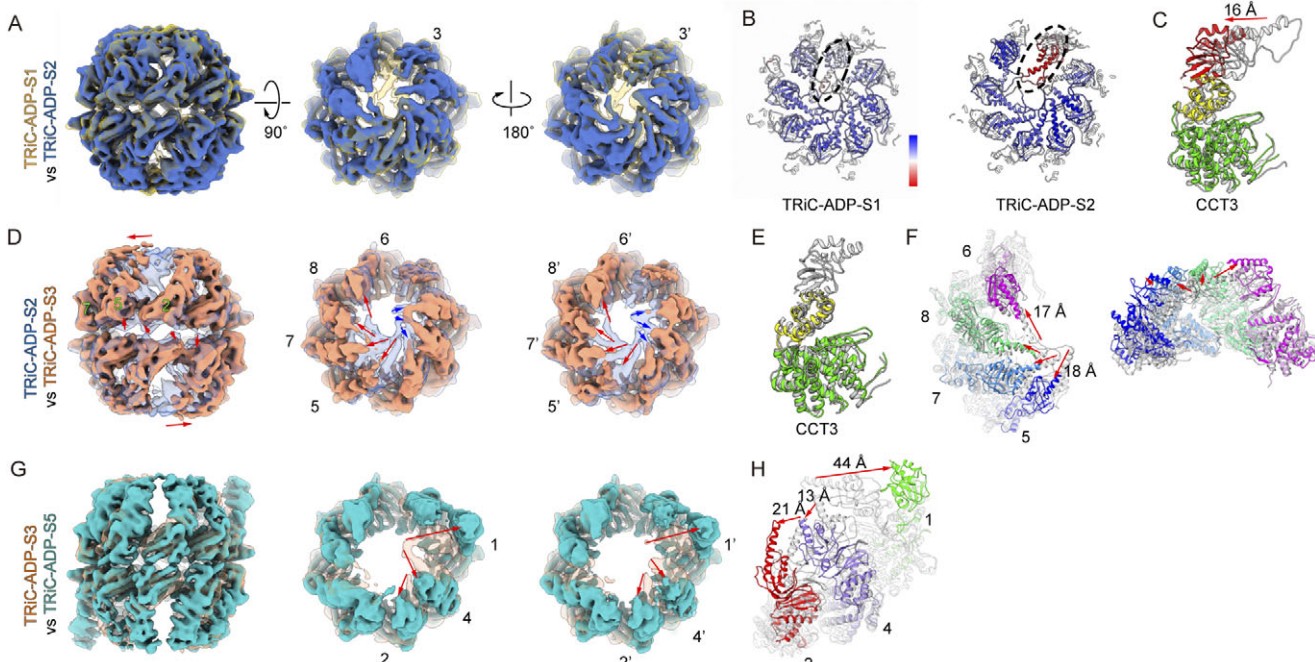

**Figure 2.** Structural comparison between TRiC maps. (A) Structural comparison between TRiC-ADP-S1 and TRiC-ADP-S2 maps reveals that in the S2 state, the CCT3 apical helical protrusion becomes disordered. Both rings exhibit similar behavior. All maps in this figure were low-pass filtered to 8 Å for structural comparison. (B) B-factor of TRiC-ADP-S1 and TRiC-ADP-S2 models. The CCT3 apical helical protrusion (indicated by dotted black ellipsoid) appears more dynamic, particularly in the S2 state, compared to other subunits. (C) Model comparison of CCT3 between S1 (in grey) and S2 (in color) states shows the A- and I-domain of CCT3 tilted outward, with the A-domain tilted by up to 16 Å. (D) Structural comparison between TRiC-ADP- S2 and TRiC-ADP-S3 maps reveals that the CCT6, CCT8, CCT7, and CCT5 subunits lean outward obviously (indicated by red arrows) in a coordinated manner. The A-domains of CCT2/CCT4/CCT1 subunits also tilt outward slightly (indicated by blue arrows). Both rings exhibit similar behavior. (E-F) Model comparison between the S2 (in grey) and S3 (in color) states shows that the CCT3 A-domain becomes disordered (E), and the A-domains of CCT6/CCT8/CCT7/CCT5 exhibit large outward leaning (F). Among them, CCT5 shows the most pronounced movement, distancing itself from the CCT2 subunit. (G) Structural comparison between TRiC-ADP-S3 and TRiC-ADP-S5 maps indicates that the CCT2, CCT4, and CCT1 subunits tilt outward to fully open the TRiC rings. (H) Model comparison between the S3 (in grey) and S5 (in color) states shows that CCT1 exhibits a tremendous outward tilting of up to 44 Å in its apical protrusion region.

In the minimally populated S4 state (1.5%), although the map resolution is not high, the trans-ring of TRiC appears to remain in a similar conformation to that in the S3 state (Supplementary Figure S4C). However, its cis-ring has opened up further, resembling the conformation seen in the fully open S5 state (Supplementary Figure S4D). We then compared the structures between the S3 and the better-resolved S5 states to capture the conformation transition in the final stage of ring opening. It appears that the A- and I-domains of CCT2 tilt up to 21 Å toward CCT5, accompanied by associated movement of CCT4 (Figure 2G, H). These collective movements of CCT2 and CCT4 greatly weaken the contacts with CCT1, particularly in the upper two domains, making it much more active. As a result, CCT1 exhibits a tremendous outward tilting of up to 44 Å, making TRiC appear wide open (Figure 2G, H, Supplementary Movies 2 and 3). Taken together, our ensemble of TRiC-ADP structures suggests that the TRiC ring-opening process occurs in a stepwise manner (Figure 2).

### CCT2 side E-domain detachment between the two rings in the S2 and S3 states

Remarkably, during the TRiC ring-opening process, especially in the transition from the S2 to S3 state, we observed a detachment between the E-domains of the two rings on the CCT2 side. As a result, a significant gap emerged between the two rings on this side (Figures 2D, 3A, and Supplementary Movie 3). This conformational transition has not been previously reported. Close

inspection of the structures revealed that during the transition from the S2 to S3 state, the CCT6 subunit drives the consecutive subunits—CCT8, CCT7, and CCT5—to expand outward as a whole (Figure 2D). This movement propagates to the distant CCT5, disrupting the interactions between the A- and I-domains of CCT5 and CCT2 (Figure 2D). Consequently, CCT5 exhibits a pronounced upward movement (Supplementary Movie 3), breaking the original E-domain interaction with CCT4' from the opposing ring (Figure 3A, and Supplementary Movie 2). Simultaneously, due to the intimate interactions between the stem-loop of CCT5 and the N- and C-termini of CCT2, CCT5 drives an obvious upward movement (up to 10 Å) of CCT2 (Figure 3A). Considering the two-fold symmetrical axis between the two rings across the CCT2-CCT2' and CCT6-CCT6' subunits, these movements collectively result in a significant gap between the equatorial domains of the two rings on the CCT2 side of the complex (between the CCT5/2/4 and CCT4'/2'/5' subunits) (Figure 3A, and Supplementary Movie 2).

Notably, in the closed S1 and the opening-initiating S2 states, the N-terminus of CCT4 bents and extends towards the outside of the ring, potentially forming contacts with the E-domains of CCT5' and CCT2' from the opposing ring (Figure 3B). Our previous structural and functional study suggested that the CCT4 N-terminus contributes to the allosteric cooperativity between the two rings (Jin et al., 2019a). During the transition to the partially open S3 state, the density corresponding to the CCT4 N-terminus is not detected (Figure 3C), most likely due to the conformational dynamics induced by the dramatic upward movement of the

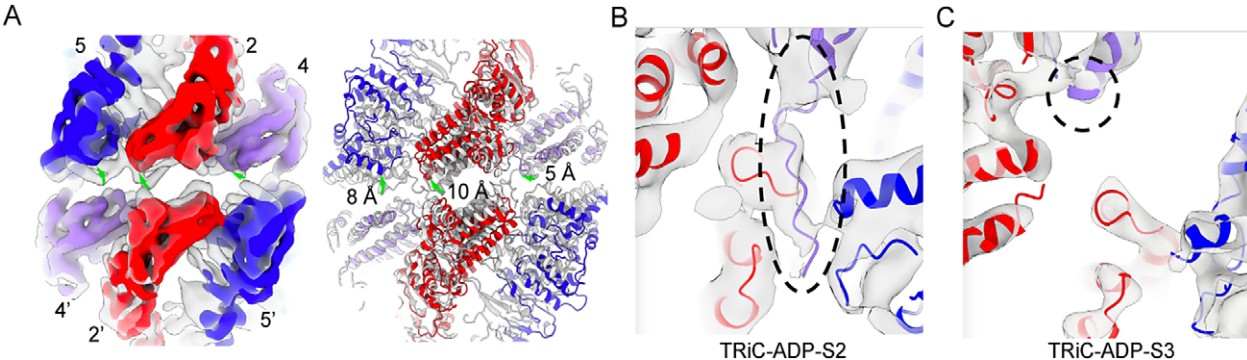

**Figure 3.** Conformational changes between TRiC-ADP-S2 and TRiC-ADP-S3 states. (A) Structural comparison between TRiC-ADP-S2 (transparent grey) and TRiC-ADP-S3 (in color) states. The view illustrates an unforeseen gap between the two rings on the CCT2 side (involving CCT5/CCT2/CCT4 and CCT4'/CCT2'/CCT5' subunits) in S3 state. The upward shifts of the E-domains from S2 to S3 states are indicated by green arrows. (B-C) A zoom-in view focusing on the N-terminal tail of CCT4, which is resolved in TRiC-ADP-S2 map (B, indicated by a black dashed oval), but appears disordered in TRiC-ADP-S3 map (C).

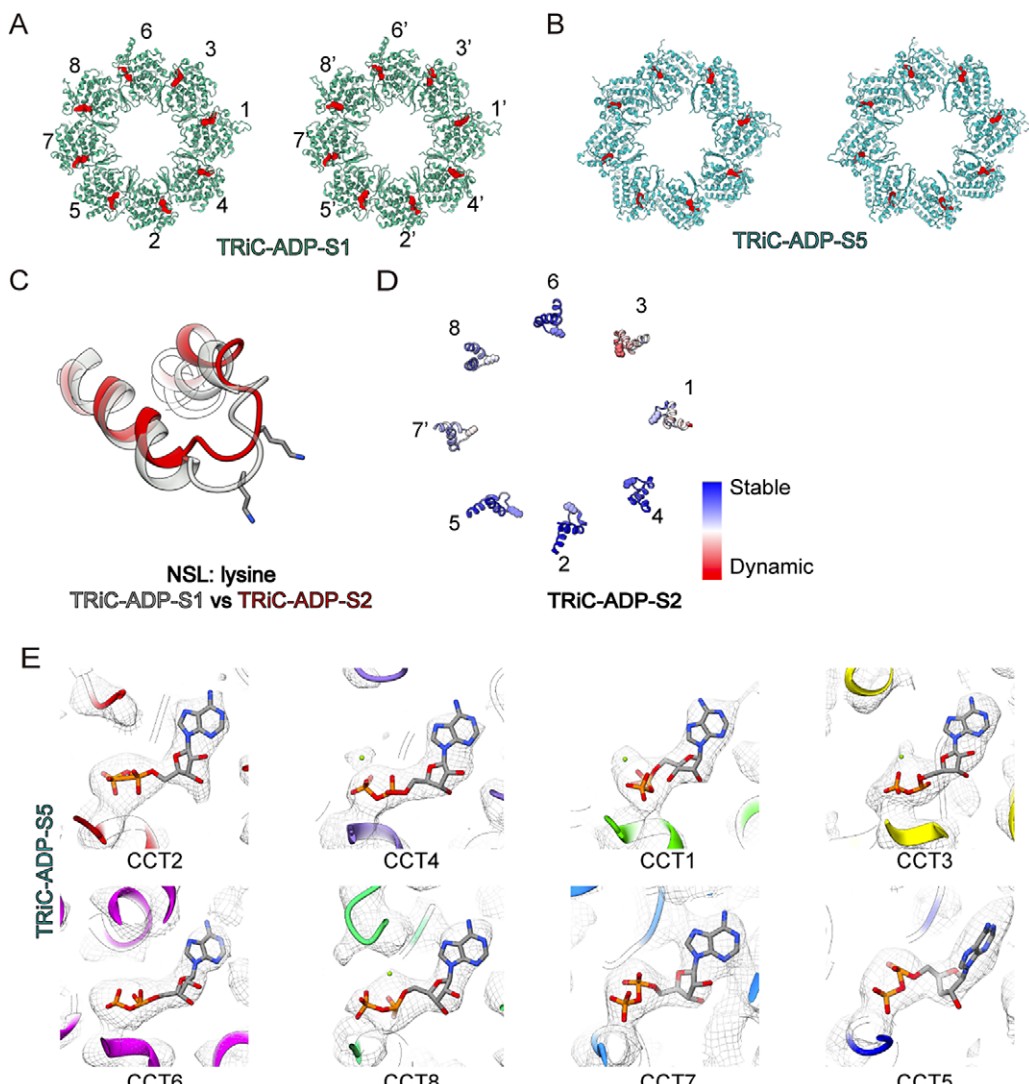

**Figure 4.** Nucleotide occupancy statuses of TRiC-ADP structures. (A-B) Nucleotide occupancy statuses of TRiC- ADP-S1 (A), and TRiC-ADP-S5 (B). Shown are slices of the model near the nucleotide pocket with nucleotide densities highlighted in red. (C) Conformational change of the NSL with the lysine on it of CCT3 from TRiC- ADP-S1 (grey) to TRiC-ADP-S2 (red). (D) In the TRiC-ADP-S2 model, the NSL of CCT3 exhibits a larger B-factor compared to other subunits, with the lysine on it being most dynamic. This suggests that the initial conformational change in CCT3 subunit may be driven by the NSL and lysine on it. (E) Zoomed-in view of the nucleotide pockets of all TRiC subunits in the TRiC-ADP-S5 state.

subunits on the CCT2 side. This indicates that the CCT4 N-terminus may retract into the inner chamber to participate in other functions, such as coordinated ring opening or substrate release.

In the subsequent S4 state, the gap between the two rings on the CCT2 side remains (Supplementary Figure S4C). Eventually, as the conformation transitions to the S5 state, the collective outward leaning of CCT5, CCT2, and CCT4 occurs, associated with the tilting of the CCT2 and CCT4 A-/I-domains towards CCT5 (Figure 2G, H). This leads to the resumption of E-domain interactions on the CCT2 side, ultimately closing the gap between the two rings on this side in the final fully open S5 state (Figure 2G and Supplementary Movie 2).

### The NSL in CCT3 detects the loss of γ-phosphate, facilitating CCT3 conformational change

We further investigated how the gradual opening of the TRiC rings is related to the nucleotide status of TRiC subunits. By examining the nucleotide-binding status of these structures (except for the TRiC-ADP-S4 state due to its lower resolution), we found that every subunit in all four structures was bound to a nucleotide (Figure 4, and Supplementary Figure S5). Since only ADP was added during incubation, we infer that all subunits are bound to ADP. This is confirmed in the relatively better resolved S1 and S5 states, which represent the initial and final stages of ring opening, where all subunits are indeed bound to ADP (Figure 4A, B, E and Supplementary Figure S5A).

The nucleotide sensing loop (NSL) and a lysine residue on it are crucial elements for sensing γ-phosphate (Pereira et al., 2012). During the ring-opening process, the NSL of each subunit gradually detects the loss of γ-phosphate, leading to a conformational change of the corresponding subunit. A conformation comparison between the S1 and S2 states reveals a most obvious upward movement in the NSL of CCT3 (up to 5.2 Å in Cα). The NSL detaches from its original position, where it typically interacts with γ-phosphate in the ATP-bound or hydrolyzing state (Jin et al., 2019a; Zang et al., 2016) (Figure 4C). This detachment could facilitate the conformational transition of CCT3, initiating TRiC ring opening. Indeed, the NSL of

CCT3 exhibits the highest B-factor in the partially open TRiC-ADP-S2 structure (Figure 4D). Collectively, our data suggest that the NSL of CCT3 detects the loss of the γ-phosphate and triggers the conformational change of the CCT3 subunit, leading to the opening of the TRiC ring. This mechanism highlights the critical role of CCT3 in the coordinated process of TRiC ring opening.

### Discussion

TRiC/CCT operates through its ATP-driven conformational cycle, converting chemical energy into mechanical forces to assist in the folding of cytosolic proteins. However, the TRiC ring-opening process and substrate release mechanism remain largely uninvestigated. In this study, by determining an ensemble of cryo-EM structures of yeast TRiC in the presence of ADP (Figure 1), we captured the TRiC ring-opening process and provided a complete picture of the conformational landscape of TRiC (Figure 2). Among the five conformational states, the TRiC-ADP-S2/S3/S4 states represent previously unobserved intermediate states in the TRiC ring-opening process (Figure 1B–D). We found that TRiC ring-opening was initiated from the dynamics of the apical-protrusion of the CCT3 subunit (Figure 2A–C), which may be related to its NSL sensitivity to the loss of γ-phosphate (Figure 4C–D). Subsequently, we observed an outward tilting of CCT6/8/7/5, followed by CCT2/4/1. There is also a detachment of the CCT2 side subunit E-domain between the two rings, which forms a gap that releases the originally bent and locked CCT4 N-terminus in this location. Finally, with the dramatic outward tilting of CCT1, TRiC fully opens both of its rings. Altogether, we captured the stepwise TRiC ring-opening process and elucidated the underlying mechanism, providing a more comprehensive picture of TRiC's conformational landscape. Our study could serve as the structural and mechanistic basis for related pharmaceutical development.

### Mechanism of TRiC ring opening

Based on our cryo-EM study on yeast TRiC in the presence of ADP and the multiple intermediate states obtained from this dataset, we propose a stepwise ring-opening mechanism of TRiC (Figure 5).

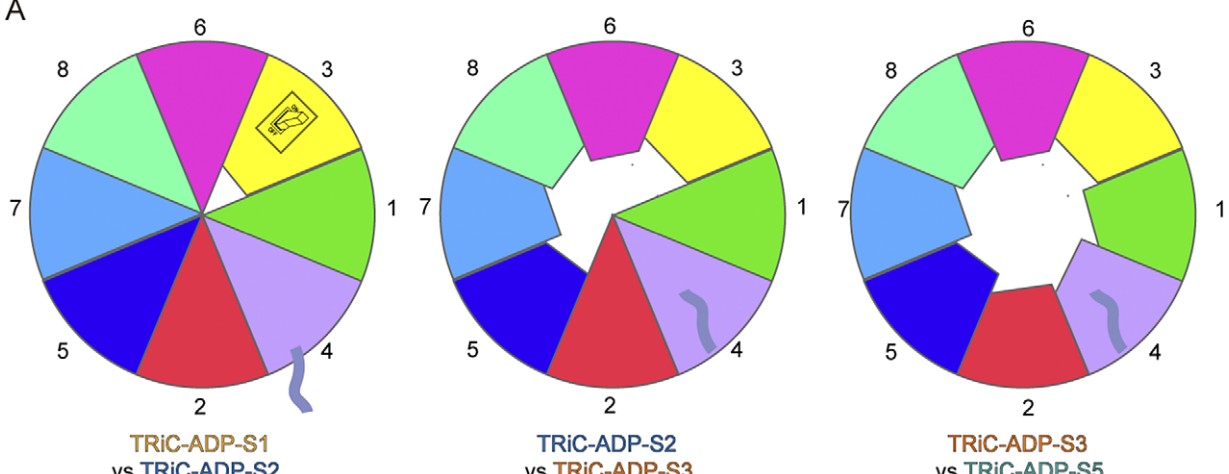

**Figure 5.** The cartoon diagram illustrates the ring-opening process of TRiC and highlights key structural elements. The extend of conformational changes in the subunits is proportional to the decreasing portions in the surrounding pie chart from TRiC-ADP-S1 to TRiC-ADP-S5. CCT3 acts as a switch, initiating the TRiC ring- opening process through the dynamics of its apical protrusion. This movement expands to the consecutive CCT6/8/7/5 subunits, followed by CCT2/4/1, ultimately leading to the opening of the TRiC rings. In the transient intermediate S2 and S3 states, a gap forms between the two rings, leading the originally bent and locked CCT4 N-terminus to retreat into the chamber.

Our findings indicate that TRiC opening is initiated by the CCT3 subunit. The NSL of CCT3 acts as a switch for the complex. When it detects the signal of γ-phosphate release, this switch is activated, triggering the apical protrusion dynamics of CCT3 in the TRiC-ADP-S2 state (Figure 5). Subsequently, CCT3 leans outward, followed by its consecutive subunits CCT6, CCT8, CCT7, and CCT5, collectively tilting outward. This movement results in the TRiC ring being halfway open in the TRiC-ADP-S3 state (Figure 1C).

Interestingly, in the very transient intermediate S2 and S3 states (sparsely populated at 6.0% and 1.5%, respectively), a gap forms between the two rings on the CCT2 side due to the detachment of the E-domains of the CCT2 side subunits (Figure 3A). This gap releases the originally bent and locked CCT4 N-terminus in this location, allowing it to retract back into the chamber and potentially participate in coordinated ring opening or substrate release (Figure 3B). These observations align with our previous study, which suggested that the CCT4 N-terminus may contribute to the allosteric cooperativity of the two rings (Jin et al., 2019a).

Subsequently, the gap on the CCT2 side closes, accompanied by the collective outward tilting of CCT2, CCT4, and CCT1 (CCT5 also involved). Notably, CCT1 exhibits the most dramatic "ejection" movement in its A- and I-domains. These coordinated movements ultimately lead to the TRiC ring becoming fully open, reaching the most populated S5 state. Overall, the opening of the TRiC rings occurs in a counter-clockwise stepwise manner, with the two rings being mostly positively coordinated.

### TRiC subunit specificity in sensing nucleotide status and substrate interactions

Interestingly, our study demonstrates that the CCT6 side subunits play a key role in initiating the ring opening and maintaining the association of the two rings during this process. Conversely, the CCT2 side subunits open at a later stage but contribute more to the allosteric coordination between the two rings. Furthermore, the obligate substrates of TRiC, including essential cytoskeleton proteins tubulin and actin, fold within the TRiC chamber by attaching to the CCT6 side subunits (primarily contacting CCT3/6/8 subunits) (Gestaut et al., 2019; Gestaut et al., 2023; Han et al., 2023; Kelly et al., 2022; Liu et al., 2023; Park et al., 2024; Wang et al., 2023). We postulate that the initial opening of the TRiC chamber on the CCT6 side could facilitate the release of folded substrates engaged with these subunits.

In addition, we previously found that in yeast TRiC, the subunits CCT3, CCT6, and CCT8 are preloaded with ADP from cellular environments in the NPP state (Zang et al., 2016). During the TRiC ring opening process, these three subunits form the initial group that detects the loss of γ-phosphate and initiates the ring opening. This finding further underscores TRiC's subunit specificity in sensing nucleotide status and substrate interactions.

### Population shifts of the conformations of TRiC in the ADP state

Based on the current study and previous reports (Cong et al., 2012; Jin et al., 2019a; Liu et al., 2023; Wang et al., 2023; Zang et al., 2016), TRiC itself may have been "encoded" to have a large conformational landscape. It has the potential to spontaneously transition among various conformations. However, many of these states are highly transient and minimally distributed, making them extremely challenging to capture. Factors such as nucleotide binding, hydrolysis, releasing, and interactions with substrates or cochaperones may shift the population distribution of TRiC's conformational states. For instance, in the nucleotide partially preloaded (NPP) state or the ATP-bound state (mimicked by AMP-PNP), TRiC remains in an open conformation, although the A-domain of CCT2 may be more stabilized in the later condition. In contrast, in the presence of ADP-AlFx (an ATP-hydrolysis transition state analog) (Jin et al., 2019a). most TRiC complexes close both rings. When TRiC interacts with substrates like tubulin, the population of the closed state can be slightly reduced (Gestaut et al., 2023; Kelly et al., 2022; Liu et al., 2023). A similar population shift occurs in other macromolecules, such as integrin, where ligand interactions are associated with global structural rearrangements involving transitions between bent, extended-closed and -open forms (Cormier et al., 2018).

In summary, in the presence of ADP, we were able to capture previously unrevealed intermediate states. These findings shed light on the TRiC ring-opening mechanism and enrich our understanding of its large conformational landscape. Furthermore, we hypothesize that the TRiC ring-closing process may share certain aspects with the revere process of ring opening. For instance, there could be a temporary detachment of the two rings at the equator on the CCT2 side to coordinate the lockage of the bent CCT4 N-terminus. However, this hypothesis requires further investigation in future studies. Overall, understanding TRiC's conformational transitions in its conformational landscape and its subunit-specific behavior is crucial for unraveling its role in protein folding and maintaining cellular hemostasis.

## Methods

### Protein purification

Yeast TRiC was purified according to our published protocol (Zang et al., 2016). The supernatant of the yeast lysate was incubated with calmodulin resin (GE Healthcare) overnight at 4 °C. TRiC was eluted using an elution buffer containing 2 mM EGTA. The pooled eluate containing TRiC was concentrated with a Millipore Ultrafree centrifugal filter device (100 kDa cutoff).

### Cryo-EM sample preparation

To prepare the TRiC sample with 1 mM ADP, the purified yeast TRiC was diluted to 1 mg/ml in the presence of 1 mM ADP and 5 mM MgCl₂ for 15 min at 30 °C prior to freezing. A 2.2 μl aliquot of this sample was applied to a glow-discharged holey carbon grid (Quantifoil, Cu R1.2/1.3, 200 mesh). To address the preferred orientation problem commonly associated with group II chaperonins, especially in the open state, the grid was pretreated with polylysine (Ding et al., 2017; Zang et al., 2016). To prepare the TRiC sample with 1 mM ATP-AlFx, the purified yeast TRiC was diluted to 1 mg/ml, and incubated in the presence of 1 mM ATP, 5 mM MgCl₂, 5 mM Al(NO₃)₃, and 30 mM NaF for 1 hr at 30 °C prior to freezing. A 2.2 μl aliquot of sample was applied to a glow-discharged holey carbon grid (Quantifoil, Cu R1.2/1.3, 200 mesh). The grid was blotted with Vitrobot Mark IV (Thermo Fisher Scientific) and then plunged into liquid ethane cooled by liquid nitrogen.

### Data acquisition

Cryo-EM movies were collected on a Titan Krios electron microscope (Thermo Fisher Scientific) operated at an accelerating voltage of 300 kV with a nominal magnification of 18,000x (yielding a pixel size of 1.318 Å after binning by 2). The movies were recorded on a

K2 Summit direct electron detector (Gatan) in the super-resolution mode under low-dose conditions in an automatic manner using SerialEM (Mastronarde, 2005). Each frame was exposed for 0.2 s, with a total accumulation time of 7.6 s, leading to a total accumulated dose of 38 e$^-$/Å$^2$ on the specimen.

### Image processing and 3D reconstruction

The image processing and reconstruction procedures and related information are shown in Supplementary Figures S1 and S2. For the TRiC-ADP sample, motion correction was performed using the embedded Motioncor2 (Zheng et al., 2017) in Relion3.0. The CTF parameters were determined using CTFFIND4 (Mindell & Grigorieff, 2003). Initially, approximately 1,000 particles were picked utilizing EMAN2 (Ludtke et al., 1999; Tang et al., 2007), and then subjected to 2D classification in Relion3.0 (Zivanov et al., 2018). Several good class averages were selected as templates for further automatic particle picking using Relion3.0. Bad particles and ice contaminations were then excluded by manual selection and 2D classification.

Good particles were used to build the initial model in Relion3.0. The entire dataset was first 3D classified into five classes. The class 5, which presented an open conformation and better structural features, was subjected to auto-refinement. CTF refinement and Bayesian polishing were applied to the refined particles. Finally, the polished particles were auto-refined, achieving a resolution of 4.05 Å after post-processing, referred to as TRiC-ADP-S5. The remaining class 3 displaying a closed conformation was further subjected to 3D classification, generating seven classes. Two of these showed distinct conformations and better structural features, leading to further refinement to obtain TRiC-ADP-S1 and TRiC-ADP-S2 maps. Another class comprising 18.0% of the particles underwent further 3D classification, resulting in four classes. Two of these displayed distinct conformations and better structural features, leading to further refinement to obtain TRiC-ADP-S3 and TRiC-ADP-S4 state maps. For the TRiC-ATP-AlFx sample, similar procedures were followed as for the TRiC-ADP sample, but only one closed conformation was detected, resulting in a 3.58 Å resolution map at closed state. The resolution estimation was based on the gold-standard Fourier Shell Correlation (FSC) 0.143 criterion, and the local resolution was estimated using Resmap (Kucukelbir et al., 2014).

### Atomic model building by flexible fitting

We built an atomic model for each of the better-resolved TRiC-ATP-AlFx, TRiC-ADP-S1, TRiC-ADP-S2, TRiC-ADP-S3, and TRiC-ADP-S5 maps. For each subunit, we selected an initial model based on its conformational similarity to our current structures, using either the closed model (PDB ID: 6KS6) (Jin et al., 2019a) or open model from TRiC-AMP-PNP (PDB ID: 5GW5) (Zang et al., 2016). These initial models were simultaneously fitted into the density map as rigid bodies using UCSF Chimera and then combined to form a complete model (Yang et al., 2012). Subsequently, *phenix.real_space_refine* was employed to improve the fitting (Adams et al., 2010). Finally, real space refinement was performed using COOT (Emsley & Cowtan, 2004) to eliminate steric clashes, followed by a last round of flexible fitting on the entire complex using Rosetta (DiMaio et al., 2015).

### B-factor calculation and rendering

We calculated B-factors for the atomic models of TRiC-ADP-S1 and TRiC-ADP-S2 using the phenix.real_space_refine script

(Adams et al., 2010). The B-factor was rendered using the command *rangecolor bfactor* in USCF Chimera.

**Open peer review.** To view the open peer review materials for this article, please visit http://doi.org/10.1017/qrd.2024.17.

**Supplementary material.** The supplementary material for this article can be found at http://doi.org/10.1017/qrd.2024.17.

**Data availability statement.** Electron density maps have been deposited in the Electron Microscopy Data Bank under accession codes EMD-45889 (TRiC-ATP-AlFx), EMD-45830 (TRiC-ADP-S1), EMD-45886 (TRiC-ADP-S2), EMD-45887 (TRiC-ADP-S3), EMD-45891 (TRiC-ADP-S4), and EMD-45888 (TRiC-ADP-S5). The associated models have been deposited in the Protein Data Bank under accession codes PDB-9CSA (TRiC-ATP-AlFx), PDB-9CR2 (TRiC-ADP-S1), PDB-9CS3 (TRiC-ADP-S2), PDB-9CS4 (TRiC-ADP-S3), and PDB-9CS6 (TRiC-ADP-S5).

**Acknowledgements.** We are grateful to the staff of the NCPSS Electron Microscopy facility, Database and Computing facility, and Protein Expression and Purification facility for instrumental support and technical assistance. This work was supported by grants from the Strategic Priority Research Program of CAS (XDB37040103), the NSFC (32130056 and 31670754), and the Shanghai Pilot Program for Basic Research from CAS (JCYJ-SHFY-2022-008).

**Author contribution.** Y.C. designed the experiments; M.J., H.W., and Y.Z. purified the proteins; Y.Z., M.J., and H.W. collected the cryo-EM data; M.J., Y.Z., and H.W. performed the cryo-EM reconstruction; M.J. did the model building; M.J. and Y.C. analyzed the data and wrote the paper with the input from Y.Z.

**Competing interest.** The authors declare that they have no conflict of interest.

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
