## [Reviewer Report]

This paper describes careful work on the dynamics of the TRiC/CCT chaperonin. The authors have carried out a cryoEM analysis of the conformational landscape of TRiC in complex with 1 mM ADP. They observe 5 structures in which the TRiC ring opens to different degrees, and they speculate that this sequence of motions is what underpins the function of TRiC as a protein folding catalyst or chaperonin. The authors have constructed a coherent story to explain their observations in a way that sheds light on the natural process involved in its function. I believe that the work is suitable for publication in QRBD, but I have the following suggestions to improve the clarity of the presentation.

Abstract – Line 8 should say “outward leaning”. Line 11should say “unforeseen temporary separation”.

An important proposal about the mechanism of ring opening is made in the paper on page 8, last 4 lines of “Results”, and on page 9 lines 3-8 of “Mechanism of TRiC ring opening”. I think this should be included in the Abstract since it is a key finding – i.e. Subunit CCT3 detects the loss of the γ-phosphate in the ATP -> ADP hydrolysis and triggers the conformational change in CCT3 leading to the opening of the TRiC ring.

On page 3, para 2, line 3, the nature of TRiC needs more introductory explanation. I suggest “… each containing eight paralogous subunits, CCT1 to CCT8, arranged …”. Please also add a sentence in the introduction to describe the percentage sequence homology between the eight subunits. Obviously TRiC experts know all this but other structural biologists need to be given this information in the introduction.

On page 4, top paragraph, there are several references that are not in the list of references at the end of the paper. These need to be corrected – Gestaut, frydman, Caixuan, Wenyu Han, wanshuxin, me, Caixuan Liu.

Page 4, last line – please explain what is ATP-AlFx. I can guess but it must be explain at first mention.

Page 5, line 3 – expand the description here to say “… inserted CBP affinity tag on CCT3 ..”

Page 6, line 11 – better to say “We therefore propose that CCT3 ..” rather than “conclude”.

Page 6, 5 lines from bottom – “weaken the constrains” -> “weaken the contacts” ?

Page 9, first 7 lines – this is a poorly explained section. Better as follows –

“… followed by CCT2/4/1. There is also a detachment of the CCT2 subunit E-domain between the two rings, which forms a gap that releases the originally bent and locked … ……. Our study could provide a structural and mechanistic basis for pharmaceutical development.”

Page 9, line 17 – better to say “sparsely” rather than “minorly.”

In the Methods section, page 13, line 7, please explain why was the dataset first classified into five classes. Why not 4 or 6? Also, on page 13, line 12, generating seven classes. Please explain why not 6 or 8.

Figure 5 – please label all 8 subunit numbers in all three parts of the diagram. At present there are only 1, 5 and 7 labelled.

---

## [Reviewer Report]

This is an excellent cryoEM paper on various states of TriC which provide insights into the mechanism of ring opening and closure during function. The analysis of the cryoEM data appears to be well done and the results sound. I recommend acceptance as is.